# A Reliable Approach for Revealing Molecular Targets in Secondary Ion Mass Spectrometry

**DOI:** 10.3390/ijms23094615

**Published:** 2022-04-21

**Authors:** Fengxia Li, Eugenio F. Fornasiero, Tal M. Dankovich, Verena Kluever, Silvio O. Rizzoli

**Affiliations:** 1Center for Biostructural Imaging of Neurodegeneration, University Medical Center Göttingen, von-Siebold-Straße 3a, 37075 Göttingen, Germany; efornas@gwdg.de (E.F.F.); tal.dankovich@med.uni-goettingen.de (T.M.D.); verena.kluever@med.uni-goettingen.de (V.K.); 2Department of Neuro- and Sensory Physiology, University Medical Center Göttingen, Humboldtallee 23, 37073 Göttingen, Germany; 3International Max Planck Research School for Neuroscience, Grisebachstr. 5, 37077 Göttingen, Germany

**Keywords:** nanoSIMS, mass spectrometry imaging, immunogold labeling, protein turnover

## Abstract

Nano secondary ion mass spectrometry (nanoSIMS) imaging is a rapidly growing field in biological sciences, which enables investigators to describe the chemical composition of cells and tissues with high resolution. One of the major challenges of nanoSIMS is to identify specific molecules or organelles, as these are not immediately recognizable in nanoSIMS and need to be revealed by SIMS-compatible probes. Few laboratories have generated such probes, and none are commercially available. To address this, we performed a systematic study of probes initially developed for electron microscopy. Relying on nanoscale SIMS, we found that antibodies coupled to 6 nm gold particles are surprisingly efficient in terms of labeling specificity while offering a reliable detection threshold. These tools enabled accurate visualization and sample analysis and were easily employed in correlating SIMS with other imaging approaches, such as fluorescence microscopy. We conclude that antibodies conjugated to moderately sized gold particles are promising tools for SIMS imaging.

## 1. Introduction

Mass spectrometry imaging (MSI) reveals the spatial variety of the chemical composition of the specimens investigated, from isotopes to macromolecular ions [1,2]. This type of information is unique and is complementary to that obtained by better-known imaging technologies such as light or electron microscopy (EM). As such, this field is rapidly growing, and applications of MSI to biological studies are thriving [3,4,5].

Several MSI platforms exist, with a variety of instrumental designs and working mechanisms. One of the most prominent platforms is secondary ion mass spectrometry (SIMS), which has shown a significant impact in biological sciences, especially due to its subcellular spatial resolution [5,6]. In SIMS analysis, a primary ion beam bombards the surface of the specimen to “sputter” a variety of secondary ions that are further extracted, focused and separated in a mass analyzer (according to their mass-to-charge ratio, m/z), before finally being detected [7,8]. Nanoscale secondary ion mass spectrometry (nanoSIMS) is a major SIMS implementation, with ionization by the orthogonal impact of highly focused reactive primary ions (i.e., Cs^+^ in positive mode or O^-^ in negative mode), which is mostly utilized for simultaneously analyzing multiple ions, which led to this technique also being referred to as multi-ion mass spectrometry (MIMS) [9,10]. It achieves a superior spatial resolution to most other MSI techniques, with pixel resolution down to under 50 nm [7]. The most common application of nanoSIMS in biology is to measure the distribution of rare stable isotopes (typically ^13^C, ^15^N, ^18^O, etc.) incorporated into newly synthesized molecules in the biological sample after pulsing the cell or animal with molecular precursors (e.g., amino acids, lipids) containing such isotopes [11]. Such experiments report the turnover of the cellular elements, such as proteins or membranes [12,13,14,15,16].

To obtain the highest amount of information from such imaging, it is desirable to differentiate subcellular structures (e.g., organelle or protein complexes) from each other within the sample. This allows localizing the sites where the stable isotope enrichment occurs and performing subsequent analyses. However, in nanoSIMS, this remains a challenge, primarily because of a lack of contrast, except for some cellular structures with obvious morphology, such as the nucleus. To overcome this difficulty, some previous studies relied on the correlation of nanoSIMS images with those obtained from other microscopy modalities (i.e., fluorescence microscopy and/or EM) to localize the region of interest (ROI) [11,16,17,18]. Nevertheless, such correlation-based methods are difficult to perform, as they require combining at least two complex instrumental modalities.

As an alternative, effort has been made to label the target organelle marker/protein with biological rare isotopes or elements that could be detected directly by nanoSIMS. One study relied on engineered peroxidase treatments to induce the labeling of the sample with osmium (Os), which is detected by nanoSIMS, albeit the treatment alters the local C, N or O levels [10]. Other approaches rely on self-synthesized probes containing fluorophores for fluorescence microscopy along with different isotopes (F, B, ^15^N) for nanoSIMS [6,19,20]. Finally, lanthanide-conjugated antibodies have been applied to simultaneously visualize multiple labels within breast cancer tissue [21]. Unfortunately, none of these probes are commercially available, implying that only the laboratories synthesizing the respective probes have, thus far, used them for experiments. This is a clear limitation of the use of labeling probes in nanoSIMS, which needs to be addressed.

In this study, we thus aimed to evaluate the commercially available probes based on gold nanoparticles (Au_Np) for preparing nanoSIMS-measurable specimens in a simple and straightforward manner, which should render nanoSIMS imaging applicable for many types of samples and experimental designs. In addition to their broad commercial availability, there are at least three advantages of using gold nanoparticles. First, 197Au is ionized well by the Cs^+^ source, which is also the source mode used for most of the biologically relevant elements (e.g., C, N, O, S, H and P). Second, gold particles are readily detected in nanoSIMS, and two studies already showed that one could, in principle, conjugate gold particles to specific antibodies [22] or nanobodies [23] to detect cellular proteins. Additionally, the last is that a broad range of commercial Au_Np conjugates are available for biological applications. Here, we thus performed a systematic study of the most popular commercially available Au_Np-conjugated labeling tools, in a variety of biological samples including human embryonic kidney 293 (HEK293) cells, neuronal cultures and mouse brain tissue. The results demonstrate the capability of the effective and specific labeling of Au_Np for nanoSIMS imaging and its great potential for biological applications. Based on our results, it is a good choice to start with 6 nm Au_Np conjugated to antibodies for biological applications in combination with nanoSIMS measurements, although other particle sizes do provide some advantages for specific experimental needs.

## 2. Results

### 2.1. Gold Particles (Au_Np) Are Readily Used in nanoSIMS

Several types of particles and conjugation modalities, as shown in Figure 1, were selected for nanoSIMS imaging in this study. These include a donkey anti-mouse secondary antibody conjugated with a very small 1.4 nm Au_Np and an Alexa546 fluorophore, and antibodies conjugated with a medium-sized 6 nm Au_Np, and also with a relatively large Au_Np, 15 nm in diameter. In addition, a 5 nm Au_Np conjugated to a His-tag binding structure (Ni-NTA-Nanogold) was also tested. Different sizes of Au_Np conjugated to secondary antibodies enable affinity labeling of target protein markers, relying on conventional primary antibodies. The anti-His-tag 5 nm Ni-NTA-Nanogold binds to His-tagged proteins, thus enabling their labeling, without any requirement for antibodies, and has been used, for example, for EM imaging [24].

To obtain a first view of the gold particles, a droplet (about 1 μL) of particle solution (6 nm Au_Np) was placed on a silicon wafer and imaged in nanoSIMS after drying. As shown in Appendix A, the particles were detected easily, and the spatial resolution reached was 212.2 ± 13.2 nm, when imaging under a 10 pA primary current. It is worth mentioning that a higher primary current comparing to that commonly applied to biological samples (i.e., ~1 pA) is not optimal for a high spatial resolution but enables a better detection, which could be one reason that the resolution calculated in this study is lower than reported earlier for Au particles imaged by nanoSIMS [16].

### 2.2. The Performance of Au_Np in Cell Culture Labeling

To test the feasibility of our workflow, secondary antibodies conjugated to Au_Np of different sizes were applied to fixed and permeabilized HEK293 cell cultures that were stained with primary antibodies labeling several organelles. For this purpose, we relied on well-known proteins that are widely used as organelle markers, including the mitochondrial import receptor subunit TOM20 homolog (TOMM20) for mitochondria, calnexin for the endoplasmic reticulum (ER), 130 kDa cis-Golgi matrix protein (GM130) for the Golgi apparatus, Lamp1 for lysosomes and 70 kDa peroxisomal membrane protein (PMP70) for peroxisome. The cartoons in Figure 2 indicate the expected pattern of each organelle marker, while the images show the nanoSIMS measurements of the 6 nm Au_Np labeling for each marker, along with the 12C14N and 32S signals from the same measurements. The features of gold signals detected for each organelle labeling are representative of the pattern expected for the respective organelles. For example, calnexin indicates relatively large, continuous structures, while Lamp1 and PMP70 show smaller, punctate patterns, and GM130 only labels specific regions in each cell.

To evaluate the labeling efficiency, a control experiment was performed without applying primary antibodies, but including the secondary antibodies carrying the Au_Np. The controls were labeled at substantially lower levels than the samples including the primary antibodies (Figure 3), confirming the specificity of the labeling procedure.

We then tested the 15 nm and 1.4 nm Au_Np-conjugated antibodies, using the same approach in cultured HEK293 cells. Their representative nanoSIMS images are shown in Appendix A, respectively. The labeling with 15 nm Au_Np generated relatively sparse Au signals for all organelle markers. Moreover, a few bright gold signal dots were also found in the controls lacking primary antibodies. The signals obtained in the experiments including primary antibodies were significantly higher than those from the control (Appendix A), albeit the large 15 nm particles seemed to prevent a thorough analysis of the morphology of the organelles, with all markers appearing equally “spotty”, which is different from the observations obtained with the 6 nm Au_Np.

Labeling with the 1.4 nm gold probe showed less intense gold signals but did reveal a characteristic pattern for each organelle type (Appendix A). The control experiment for the 1.4 nm gold labeling had very low signals (Appendix A), again confirming that these particles provide specific labeling, in spite of the fact that the signals obtained were lower than those provided by the 6 nm Au_Np (as observed by comparing Figure 3a and Appendix A).

Overall, these experiments indicate that the use of Au_Np conjugated to secondary antibodies is possible for different particle sizes, albeit 1.4 nm gold particles provide relatively few counts and as such lead to dim images, while the 15 nm particles do not seem to reveal all epitopes, resulting in “spotty” images. To test whether the 5 nm Ni-NTA-Nanogold, which targets His-tagged molecules, is also usable in similar analyses, we employed a recombinantly produced, His-tagged protein of the extracellular matrix, Tenascin R (TNR). This protein was added to neuronal cultures, where, as the endogenous TNR, it integrates in the extracellular matrix and is readily detected by, for example, anti-His-tag probes [25]. However, the 5 nm Ni-NTA-Nanogold particles were unable to detect it, since they resulted in very high unspecific labeling also in control samples, lacking any His-tagged proteins (Appendix A). While this type of experiment may be successful with substantial optimizations, we conclude that it is less facile than antibody-based experiments, and that it would therefore be a less preferred option for broader biological applications.

### 2.3. Au_Np Reveals Its Targets in Tissues

To assess if the gold particle-based probes could be used in tissues, we relied on the 6 nm Au_Np conjugated to donkey anti-mouse antibodies, labeling the same organelle markers in wild-type mouse brain tissue slices. The labeling procedure was similar to that of the HEK293 cell labeling, with some fine tuning of the permeabilization, blocking and washing steps (see Methods for details). An overview of representative mouse brain slice images for each organelle is shown in Appendix A. In the control staining, hardly any gold signal could be detected, while bright gold signals were measured for all organelle markers. Unlike the more continuous patterns detected in cultured cells, the slices exhibited a “spotty” pattern, which, however, is expected, due to the less efficient penetration of both primary and secondary antibodies in the brain slices [18].

During the labeling optimization, we also sought to confirm the Au_Np labeling by a comparison to fluorescence imaging, through detecting primary antibodies by both Au_Np (6 nm)- and fluorophore-conjugated secondary antibodies, relying on an Atto542-conjugated secondary nanobody (Figure 4). Most of the strong gold signals correlated well with the fluorescence (Cyanine 3: Cy3) signal (Figure 4a,b). Overall, the signals correlated significantly in all samples we analyzed (Figure 4c), which suggests that the Au_Np labeling is indeed specific.

Finally, to employ the Au_Np labeling for a relevant biological experiment, we tested the brain slice labeling using brains from mice that were fed with ^13^C_6_ L-lysine for 14 or 21 days. This results in an enrichment of ^13^C in their newly synthesized proteins, which is expected to reach significant, measurable values [12]. We then analyzed the ratio of ^13^C to ^12^C, as an indication of protein turnover, in areas labeled by Au_Np, targeting Lamp1, calnexin, GM130 and PMP70 (Figure 5). Interestingly, the Lamp1-marked lysosomes contained significantly lower levels of ^13^C than the rest of the cell, which is in line with these lysosomes being degradation machineries that would collect and process selectively old proteins [26]. A similar effect was seen for calnexin, which is in line with our previous observation that the ER is, in relative terms, an old organelle in neuronal cells [16]. The Golgi apparatus, revealed by GM130, was not significantly different from the rest of the cell. Finally, the PMP70-marked peroxisomes contained more ^13^C than the rest of the cell, which is in good agreement with the very short lifetime of this protein in the brain [12], suggesting that peroxisomes are short-lived and are rapidly replaced in this tissue.

## 3. Discussion

Our systematic study on gold particle labeling of organelle markers for nanoSIMS imaging demonstrated that gold nanoparticles can be used to label a broad range of biological targets in samples from cancer cell cultures to brain tissues, with high specificity. With regard to the performance of various alternative sizes of Au_Np, this study indicates that the 6 nm particle size is a good starting probe when designing Au_Np-based tagging of molecules for nanoSIMS imaging. The 1.4 nm particle labeling has the disadvantage of a relatively low sensitivity (low signals), although it also performs well with regard to the labeling of the representative structure of the target. Moreover, its shelf life is only one third of that of the 6 nm- and 15 nm-conjugated probes, according to the manufacturers. Nevertheless, probes with 1.4 nm Au_Np could be useful in studies that have limited permeabilization possibilities, since they should penetrate tissues almost as well as unlabeled antibodies. Regarding the 15 nm particle-conjugated probes, although they can be imaged well, they provide a less representative picture of their targets comparing to the 6 nm- and 1.4 nm-conjugated probes (Appendix A vs. Figure 2 and Appendix A). This is probably due to their relatively large particle size and lower penetration capability, which results in less dense labeling with respect to Au_Np of a smaller size. An additional observation on the 15 nm gold labeling is that the control also showed bright gold signal spots, albeit not very numerous, which could limits its applications. Due to their large particle size (Figure 1), it is expected that the unbound particles were not efficiently washed from the samples. This might also generate high signals, which implies that stronger washing and/or permeabilization measures should be taken for removing larger probes. Overall, this implies that the 6 nm Au_Np is an optimal starting point for conventional experiments. Some degree of sample preparation tuning should be expected with regard to each specific application, as we also realized using the similarly sized 5 nm Ni-NTA-Nanogold probe, which failed to differentiate positive labeling and controls, and in which the washing procedures probably need to be vastly improved.

These commercially available Au_Nps are more user-friendly than most probes containing rare elements (i.e., ^19^F, boron) that were used in previous nanoSIMS works. As an example, previous work from our group by Vreja et al. [6] described a labeling strategy using the self-synthesized probe SK155 that contains ^19^F as an isotopic label for nanoSIMS, an Abberior Star 635 fluorescent moiety for fluorescence imaging and a reactive azide group for linking to alkyne containing an unnatural amino acid, which needs to be incorporated into the cell by genetic encoding to express the target protein, together with the respective tRNA and aminoacyl–tRNAsynthetase, followed by click chemistry reactions. The authors tested, among others, molecules targeted to the endoplasmic reticulum, which resulted in broadly similar patterns to our present observations, albeit at a higher density, due to the use of smaller labeling probes, and a high protein overexpression rate. Following a similar strategy, Kabatas et al. [19] further synthesized the similar probe BorEncode containing boron for nanoSIMS imaging in the positive ion mode. They then achieved the visualization of newly synthesized proteins that incorporated homopropargyl-l-glycine (HPG) targeted by this probe with click chemistry reactions. Moreover, a second probe, Borlink, was also produced, which allowed the conjugation of this probe to nanobodies, for affinity labeling, as was also later performed for ^19^F to be incorporated in affinity probes [20]. These probes were used to reveal mitochondria or peroxisomes, as in our current work, with comparable results to our findings. Overall, these previously published probes are smaller than the antibodies used here and therefore offer the promise of denser labeling, but they all need to be self-synthesized, and the click-chemistry-based probes are extremely difficult to apply to cells. A potential alternative is the use of rare metals coupled to antibodies, as conducted in a study that detected 10 labels within breast cancer tissue simultaneously [21]. This approach is similar to the one we employed here but again relies on coupling reactions that need to be self-performed.

Previous attempts have also been made to utilize gold [22,23], which could be ionized and detected well in nanoSIMS. In those studies, however, more efforts were also devoted to the self-conjugation of gold to immunostaining molecular targets, for example, by conjugating 1.4 nm Au_Np to antibodies for labeling actin in mouse intestinal cells [22]. Moreover, they also labeled Ribeye and synaptophysin with 1.4 nm fluoronanogold-Fab’ conjugates. The procedure they used, labeling the samples after plastic embedding and ultrathin processing, is difficult to apply for most users and tends to fail for unabundant proteins or non-ideal antibodies. In previous work, we tested tens of antibodies for labeling after plastic embedding and found most to result in labeling close to background levels, which implies that the use of conventional labeling methods, as performed in this study, is a more desirable alternative. Another interesting recent study of gold nanoparticles focused on conjugating 3 nm Au_Np to secondary nanobodies (nanobody–gold conjugates) [23]. Nanobodies have the advantage of a much smaller size (~2-3 nm) in comparison to the antibodies used in the current study, which potentially leads to a higher spatial resolution [27,28]. However, these small probes of nanobody–gold conjugates are stabilized by disulfide bonds [29], which may adsorb onto the gold surface [30], causing the nanobody to lose its conformation. This implies that such tools may have a shorter shelf life than that of the larger antibodies, and potentially some variation from batch to batch. The 6 nm gold particle conjugated to a normal antibody applied in the current study has a (company-verified) longer shelf life, which should help with experimental reproducibility. In addition, as for the other probes mentioned above, the nanobody–gold conjugates are not commercially available. Therefore, in comparison to the above-mentioned Au_Np- or other element-based probe studied, the commercially available Au_Np-based probes evaluated here have the advantage of direct and easy access to all the potential users that can conveniently utilize them with their conventional labeling experience, for a broad range of biological materials and with flexible experiment setups. These are therefore a very user-friendly set of probes for nanoSIMS.

With regard to biological applications, one limitation of previous studies is that they did not combine probe labeling with the analysis of metabolic or cellular activity. To perform this here, we used the 6 nm Au_Np labeling to test protein turnover in vivo. By identifying the organelles through the visualization of gold signals, we could, for example, detect the accumulation of aged proteins in the lysosome region (Figure 5). This finding from the direct nanoSIMS labeling is very promising. Based on similar concepts and strategies, lipid and carbohydrate metabolism and degradation could be investigated as well, expanding the utility of our workflow. Au_Np-based constructs may also be studied by other mass spectrometry imaging instruments (such as laser ablation inductively coupled plasma mass spectrometry (LA-ICPMS) and time-of-flight SIMS (TOF-SIMS)), in order to obtain even more information on the sample material found in the vicinity of the gold particles [31,32,33,34,35,36]. Additionally, the gold particle labeling provides valuable information for correlating fluorescence images with nanoSIMS images. As shown in our correlative experiment (Figure 4), targeting the same molecule with both a gold probe and a second fluorophore probe allows matching the two images with high precision. Taking advantage of this approach, additional fluorescence channels could be used for localizing multiple organelles in nanoSIMS images.

Despite some obvious advantages, there are also several limitations of using Au_Np-based probes. Relatively big gold nanoparticle conjugation to full antibodies leads to a fairly large sized probe (Figure 1), which potentially decreases the penetration ability, resulting in potentially limited epitope detection. Particularly in tissue staining, sufficient permeabilization and an increased incubation time with the labeling solution are critical for good performance. However, this also has a positive aspect, since the addition of very dense probes can lead to a diluting effect, by incorporating large amounts of probe atoms into the samples (i.e., C, N, H, S and O) and competing away the sample material that consisted of stable isotopes (^13^C, ^15^N, ^2^H, etc.) in the respective area. This was, for example, an effect in a study using OsO_4_ labeling of the samples [10]. Overall, we conclude that the Au_Np probes are valuable tools for SIMS imaging, which should replace some fluorescence-SIMS correlation-based methods and should make SIMS applications in biology more user-friendly.

## 4. Materials and Methods

### 4.1. The Materials and Antibodies

The materials used in this study are listed in Table 1. The antibodies and gold nanoparticles applied can be found in Table 2.

### 4.2. Cell Culture

#### 4.2.1. HEK293 Cell Culture

Human embryonal kidney HEK293 cells were obtained from the German collection of microorganisms and cell cultures (DSMZ ACC 305), and their culture was performed in Dulbecco’s modified eagle medium (DMEM) mixed with 10% fetal calf serum (FCS), 4 mM L-glutamine and 100 U/mL penicillin and streptomycin. Before each labeling experiment, cells were seeded on poly-L-lysine (PLL)-coated coverslips in a 24- or 12-well plate and incubated at 37 °C and 5% CO_2_ overnight to be ready for use.

#### 4.2.2. Neuronal Culture

Details about the neuron culture procedure were described in previous publications [37]. Briefly, hippocampal neuronal cultures were prepared from dissected hippocampi of newborn Wistar rat pups (P0). After dissociation, they were seeded on PLL-coated glass coverslips and were allowed to adhere to the coverslips. They were subsequently incubated in Neurobasal-A medium (Life Technologies, Carlsbad, CA, USA) at 37 °C and 5% CO2 for 14–16 days before use.

### 4.3. Tissue Preparation

Details about the tissue preparation can be found in a previous study [25]. Mouse brain tissue was dissected, snap frozen (−80 °C) and stored before use. A thick slice was then cut from it and fixed overnight at 4 °C in 4% paraformaldehyde (PFA). It was subsequently embedded using Tissue- Tek^®^ O.C.T.™ Compound (Sakura, Finetek USA Inc., Torrance, CA, USA) and further stored at −80 °C until it was sectioned into ~30–40 µm-thick slices using a Leica CM1850 cryotome and placed in phosphate-buffered saline (PBS) ready for immunostaining. Both brains from wild-type mice and ^13^C_6_-lysine-pulsed mice were treated in the same way as mentioned above. Pulsing of ^13^C_6_-lysine was conducted as previously described [12]. Briefly, wild-type mice were fed with ^13^C_6_-lysine containing food for 14 or 21 days before they were sacrificed. All mouse experiments were approved by the local authority, the Lower Saxony State Office for Consumer Protection and Food Safety (Niedersächsisches Landesamt für Verbraucherschutz und Lebensmittelsicherheit), as also specified in the Institutional Review Board Statement.

### 4.4. Immunostaining

#### 4.4.1. Staining of HEK293 Cells

HEK293 cell cultures were first fixed in 4% PFA in PBS for 10 min on ice followed by 20 min at room temperature (RT). They were then quenched with 100 mM glycine for 20 min. Afterwards, permeabilization was performed in PBS solution containing 0.1% Triton X (#9005-64-5, Merck, Germany) for 30 min. The cells were then incubated overnight at 4 °C with one of the organelle-targeting primary antibodies, namely, TOMM20 (1:20), PMP70 (1:400), Calnexin (1:100), GM130 (1:50) and lamp1 (1:50), diluted in blocking solution. Detailed information on primary and secondary antibodies is listed in Table 2. Depending on the primary antibodies applied, the blocking solution was either PBS containing 3% tryptone and 0.1% Triton X for GM130 and Lamp1, or PBS containing 2% BSA (A1391- 0250; Applichem, Germany) and 0.1% Triton X for TOMM20, PMP70 and Calnexin. After primary antibody incubation, samples were washed 3 times for 5 min each with blocking solution. Except for TOMM20 as the primary antibody, they were subsequently incubated with secondary antibodies of goat anti-rabbit IgG (1:100) for 2.5 h at RT. The washing was repeated and then followed by further incubation with secondary antibodies of mouse anti-goat IgG (1:100) for 2.5 h. After that, donkey anti-mouse IgG conjugated to Au_Np with a size of 6 nm (1:20) or 15 nm (1:20), or goat anti-mouse IgG conjugated to 1.4 nm Au_Np and Alexa Fluor^®^ 546 (1:30) was applied as the last staining antibody. For targeting mouse anti-TOMM20, Au_Np-conjugated anti-mouse secondaries were applied directly after applying the primary antibody, without signal amplification with goat anti-rabbit IgG and mouse anti-goat IgG. Samples were subsequently washed with high-salt PBS (PBS + 350 mM NaCl) 3 times for 5 min each and PBS 3 times for 5 min each. Before embedding, the samples were post-fixed with 4% PFA mixed with 1% Glutaraldehyde for 20 min, washed with PBS, quenched with 100 mM glycine for 15 min and eventually washed twice with PBS.

#### 4.4.2. Labeling of His-Tagged TNR

Labeling of His-tagged TNR was performed in live neurons by applying 5 µg/mL recombinant TNR (# 3865-TR; Biotechne GmbH, Germany) together with Ni-NTA-Nanogold (1:25), diluted in neuronal culture media for 1 h at 37 °C. As a control, neurons were incubated with Ni-NTA-Nanogold diluted in culture media alone. The neurons were then washed 3 times with Tyrode solution (124 mM NaCl, 5 mM KCl, 2 mM CaCl_2_, 1 mM MgCl_2_, 30 mM glucose, 25 mM HEPES, pH 7.4). Afterwards, they were fixed with 4% PFA and 1% Glutaraldehyde for 20 min, followed by quenching with 100 mM glycine for 15 min. They were embedded and sliced before nanoSIMS measurement following the same process as described in the following Section 4.5.

#### 4.4.3. Staining of Brain Tissue Slices

Immunostaining of brain tissue slices basically followed a similar procedure to the cellular culture staining, but with some fine tuning, which was based on the pre-experiment performed with fluorophore-conjugated secondary antibody staining. The 4% PFA fixation and its corresponding quenching step were not conducted for brain slices because fixation was already performed before sectioning, as described in the tissue preparation, Section 4.3. The staining procedure began with 40 min of permeabilization/blocking with staining solution, i.e., PBS containing 0.5% Triton X, and 3% BSA or 3% tryptone depending on the primary antibody applied, as described for the cellular culture staining. After permeabilization, the sequential staining process of primary and secondary antibodies was the same as for the HEK293 cell staining, but with the incubation conducted by placing brain slices in staining solution instead of coverslips on top of an antibody solution, and at 50 RPM for secondary antibodies. Moreover, the washing step between two antibody applications for brain slices lasted longer, specifically 3 times for 10 min each at 50 rpm. After staining and before post-fixation, the samples were kept in PBS overnight to allow a thorough wash-up. At last, the post-fixation was conducted using a 4% PFA and 1.5% glutaraldehyde mixture for 30 min followed by quenching with 100 mM glycine together with 100 mM NH4Cl for 20 min.

For the correlative experiment, the staining of brain slices was only different in the step of applying the Au_Np-conjugated secondary antibody. To be specific, the 6 nm Au_Np conjugated to donkey anti-mouse antibody was mixed with Atto542-conjugated nanobody (1:100) and applied together to the sample.

### 4.5. Embedding and Slicing

After immunostaining, samples were embedded in medium-grade LR white (London Resin Company, London, UK) using a similar procedure to that detailed in a previous study [38]. For cell cultures, samples were partially dehydrated at first in 30% ethanol for 10 min followed by 3 times dehydration in 50% ethanol for 10 min each at 50 rpm. Afterwards, the samples were incubated for 1 h with a 1:1 mixture of 50% ethanol (in ddH_2_O) and LR white, and then for 1 h in pure LR white at 50 rpm. Following LR white incubation, the coverslips were transferred onto a pre-cooled metal plate, and the samples were incubated in a mixture of LR white and LR white accelerator at RT to allow for initial hardening/sealing. Afterwards, LR white and LR white accelerator were freshly mixed and added to the sample for finalizing the embedding through incubation at 60 °C for 90 min. Samples were allowed to cool down after oven incubation and then cut into ~200 nm-thin slices on an ultramicrotome (EM UC6, Leica Microsystems, Wetzlar, Germany) to be placed onto silicon wafers (Siegert Wafer GmbH, Aachen, Germany) for further imaging. For tissue samples, the embedding and slicing procedure was similar but with stronger dehydration. The consecutive dehydration steps were as follows: brain slices were dehydrated for 10 min in 30% ethanol, 10 min in 50% ethanol and then 3 times of 10 min each in 70% ethanol at 50 rpm. They were then incubated for 1 h in the 1:1 mixture of LR white and 70% ethanol (in ddH_2_0). The procedure afterwards was basically the same as for the cell cultures.

### 4.6. Fluorescence Imaging

For brain slices labeled both with the gold-conjugated antibody and fluorophore-conjugated nanobody, fluorescence imaging by fluorescence microscopy was conducted after tissue embedding and slicing on silicon wafers to localize the area of interest for further nanoSIMS imaging and correlation. Details on the imaging can be found in a previous study [38]. Briefly, fluorescence imaging was performed on an inverted Nikon Ti-E epifluorescence microscope (Nikon Corporation, Chiyoda, Tokyo, Japan) operated via the Nikon NIS Elements AR software (version 4.20; Nikon). A Plan Apochromat 100×, 1.59 NA oil immersion objective and a 1.5× optovar lens were applied to achieve a pixel size of 106 × 106 nm.

### 4.7. Correlation Procedure

The overlay of fluorescence images and nanoSIMS images from the correlative experiment was conducted in Photoshop. The histological overlay of the biological area from two instrumental measurements was based on green fluorescent channel and 12C14N images from fluorescence and nanoSIMS imaging, respectively. Fine tuning of the more accurate image overlay was based on the Cy3 and 197Au signals. During this process, Cy3 signal images were warped in Photoshop when it was necessary to correct for distortions caused by the high vacuum of the nanoSIMS [38].

### 4.8. SIMS Imaging

SIMS images were taken by a NanoSIMS 50 L instrument (Cameca, France) in negative ion mode with an 8 kV Cs^+^ primary ion source. To reach the steady state of ionization, implantation of the targeting imaging area was performed prior to taking each image. This was conducted at a primary ion current of ~135 pA for about 10–15 min. Tissues were generally implanted for longer than cell cultures. An entrance slit (ES3) and an aperture slit (AS1) were applied for separating isobaric mass peaks with a sufficient mass resolution. Sample widths of 12–40 µm were imaged with 128 × 128 pixels or 256 × 256 pixels using a primary ion current of 10 pA at D1-3. Images of 12C14N^-^, 32S^-^ and 197Au^-^ were taken for both cell culture and tissue samples, and images of 12C15N^-^ or 13C14N^-^ were additionally taken for brain tissue samples, which were named 12C14N, 32S, 197Au, 12C15N and 13C14N in the paper. The dwell time of the primary ion beam was adjusted for different ions since their abundance varies a lot. However, 12C14N images and 13C14N images from brain slices were always taken simultaneously under the same magnetic field with the same dwell time to generate isotopic ratio calculations that are as accurate as possible. A dried 6 nm gold standard drop on a silicon wafer was imaged basically with the same instrumental setup as for the real sample, while a few analysis parameters were set differently accordingly. The implantation time was shorter by about 2–3 min. Additionally, the pixel resolution was higher with a 7 × 7 µm area scanned by 512 × 512 pixels.

### 4.9. Analysis Routine

NanoSIMS images were first opened through the OpenMIMS plugin, ImageJ (NRIMS, Cambridge, MA, USA), and then exported in a variety of image types, which were required for further processing. To calculate the spatial resolution of gold standard particle imaging, a line scan was drawn through the gold spot and fitted with a Gaussian curve to calculate the full width at half maximum (FWHM). To compare the labeling intensity, cellular average 197Au and 12C14N counts were computed for 4–10 cells per condition. For calculating the correlation coefficient of 197Au images with fluorescence images, we drew lines through manually selected spots, across both the SIMS and fluorescence images, before calculating Pearson’s correlation coefficient for the respective line scans. The line scans were also drawn across the same areas in mirrored fluorescence images, to detect the random colocalization of the gold signals to fluorescence signals. For investigating the protein turnover in different organelle-marker-labeled areas, we selected gold-containing or gold-lacking areas by manually drawing area outlines, followed by a calculation of the required isotopic ratios. These analyses were performed using self-written routines in Matlab (Mathworks, Inc., Natick, MA, USA).

### 4.10. Statistics

Correlations were calculated as Pearson’s coefficients in MATLAB. Statistical significance was calculated by the Wilcoxon rank sum test in R. A *p*-value of < 0.05 was defined as statistically significant. Boxplots comprised the following: mid-line = median; boxes = 50th percentile; error bars = 75th percentile; black dots = outliers; orange dot = mean.

## Figures and Tables

**Figure 1 ijms-23-04615-f001:**
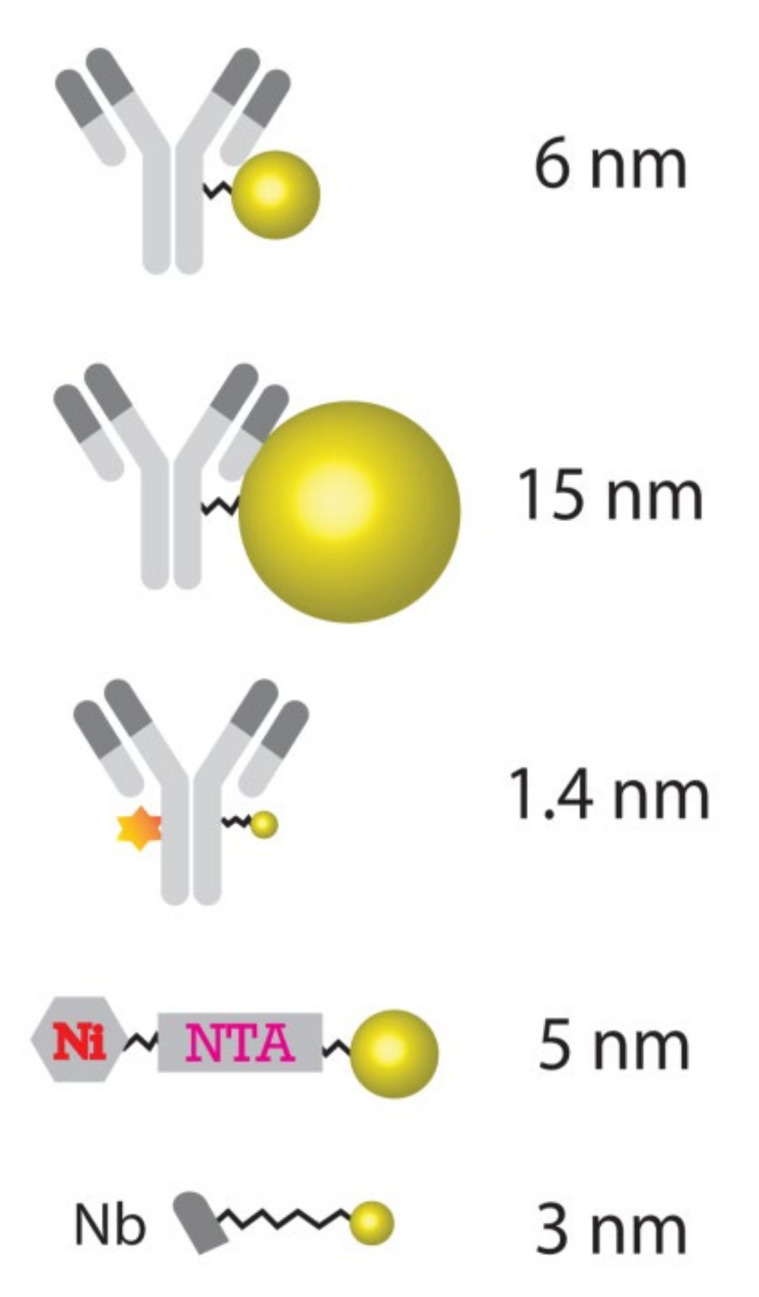
Schema of Au-based probes measured by nanoSIMS. Secondary antibodies were conjugated to Au_Np (ball) with a size of 6 nm, 15 nm and 1.4 nm, and the 1.4 nm-conjugated antibody was additionally conjugated to a fluorophore (star). The nanobody (Nb) was conjugated to 3 nm Au_Np. All of them enable the affinity labeling of target molecules. The 5 nm Au_Np can be attached to His-tagged proteins through the binding of its conjugated nickel-NTA to His-tags.

**Figure 2 ijms-23-04615-f002:**
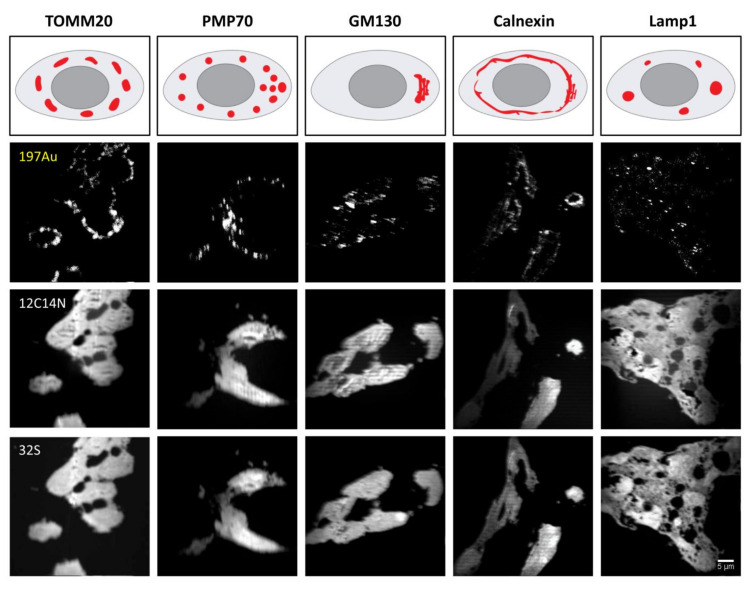
NanoSIMS detection of 6 nm Au_Np-tagged HEK293 organelle markers. Schematic representation of the expected organelle distribution from thin tissue slices after embedding and sectioning. The 197Au signal from antibody labeling represents the specific targeted protein, while the 12C14N and 32S signals reveal the cells. The signals are produced by a broad range of biological materials (proteins, DNAs, RNAs, etc.) that contain a substantial amount of C, N and S elements. One representative nanoSIMS image from each organelle labeling is shown.

**Figure 3 ijms-23-04615-f003:**
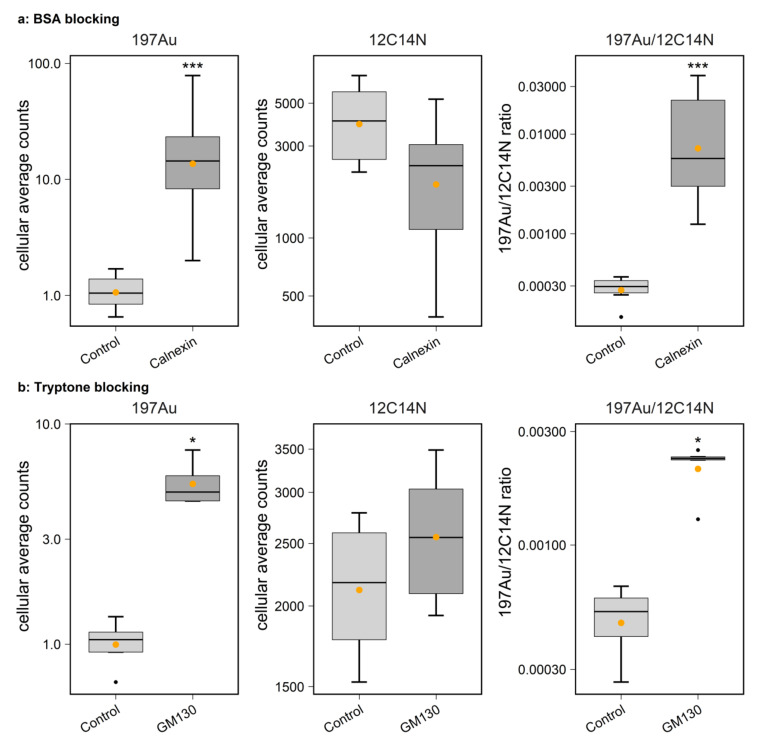
Boxplots show the 6 nm Au_Np labeling intensity of HEK293 organelle markers in comparison to the control in which no primary antibodies were applied. A significantly higher cellular labeling intensity was detected for both (**a**) bovine serum albumin (BSA) blocking (Wilcoxon rank sum test, N = 6 and 10 for control and calnexin, respectively; *p* = 2.5 × 10^−4^, 0.073 and 2.5 × 10^−4^ for 197Au, 12C14N and 197Au/12C14N, respectively) and (**b**) tryptone blocking (Wilcoxon rank sum test, N = 4 and 5 for control and GM130, respectively; *p* = 0.016, 0.29 and 0.016 for 197Au, 12C14N and 197Au/12C14N, respectively). * *p* ≤ 0.05, *** *p* ≤ 0.001.

**Figure 4 ijms-23-04615-f004:**
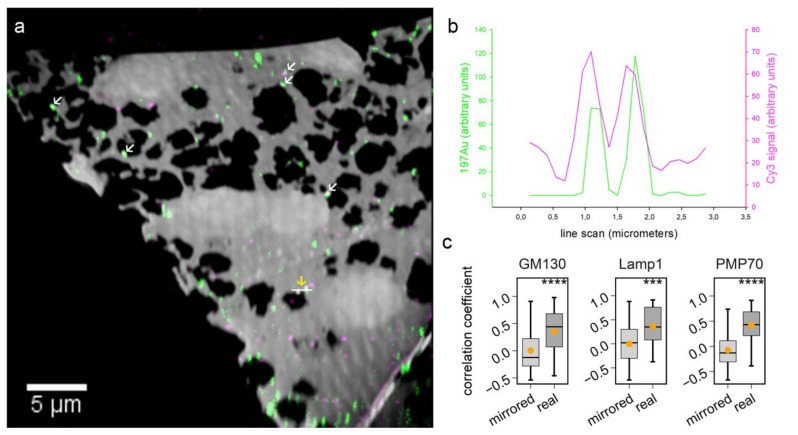
Correlation of 6 nm Au_Np detected by nanoSIMS and fluorescence signals detected by fluorescent microscopy. (**a**) Overlay of gold images and fluorescent images for targeting PMP70. Examples of colocalized gold and fluorescent signals are indicated by white arrows. The signal variations of gold and Cy3 along the area indicated by a white horizontal line were extracted and plotted in (**b**). (**c**) shows the correlation coefficient between gold and fluorescent images (real) in comparison to that between gold and mirrored fluorescence images (mirrored). Wilcoxon rank sum test: for GM130, lamp1 and PMP70, N = 51, 45 and 50, respectively, and *p* = 4.9 × 10^−5^, 1.3 × 10^−4^ and 1.9 × 10^−9^, respectively. *** *p* ≤ 0.001, **** *p* ≤ 0.0001.

**Figure 5 ijms-23-04615-f005:**
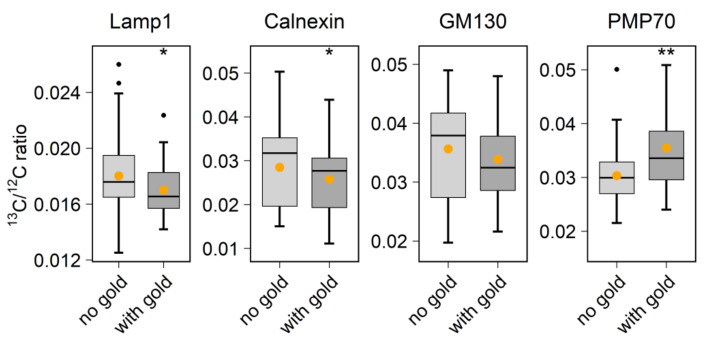
Detection of protein turnover heterogeneity in different organelles of ^13^C isotopically enriched mouse brain slices. For each organelle marker staining, ^13^C/^12^C ratios were calculated for 6 nm Au_Np-labeled areas (with gold) and the random selection of multiple representative unlabeled areas (no gold). A difference in the relative protein “age” in different organelles, in comparison to the average cellular protein “age”, is demonstrated by this approach. Wilcoxon rank sum test: for lamp1, calnexin, GM130 and PMP70 (no gold, with gold), N = (113, 41), (59, 80), (56, 40) and (47, 34), respectively, and *p* = 0.019, 0.044, 0.28 and 0.004, respectively. * *p* ≤ 0.05, ** *p* ≤ 0.01.

**Table 1 ijms-23-04615-t001:** Materials used for experiments.

Chemicals, Recombinant Proteins	Source	Identifier
LR White medium grade	Agar Scientific	Cat#: AGR1281
Accelerator for LR White resin	Agar Scientific	Cat#: AGR1281
Recombinant Tenascin-R	Biotechne GmbH	Cat#: 3865-TR

**Table 2 ijms-23-04615-t002:** Antibodies and gold particle materials applied.

Antibodies and Gold Particles	Source	Identifier
Rabbit polyclonal anti-Calnexin	Abcam	Cat#: ab22595
Rabbit polyclonal anti-PMP70	Abcam	Cat#: ab85550
Rabbit polyclonal anti-lamp1	Abcam	Cat#: ab24170
Rabbit polyclonal anti-GM130	Sigma	Cat#: G7295
Mouse monoclonal anti-TOMM20	Sigma	Cat#: WH0009804M1
Goat Immunoglobulin G (IgG) anti-rabbit IgG	Dianova	Cat#: 111-005-144
Mouse anti-goat IgG	Dianova	Cat#: SBA-6158-01
Donkey-anti-Mouse IgG—6 nm Gold	Aurion	Cat#: 806.322
Donkey-anti-Mouse IgG—15 nm Gold	Aurion	Cat#: 815.322
Alexa Fluor^®^ 546 FluoroNanogold™ IgG Goat anti-Mouse IgG	nanoprobes	Cat#: 7401
5 nm Ni-NTA-Nanogold	nanoprobes	Cat#: 2082
Atto542 conjugated anti-mouse nanobody	nanoTag	Cat#: N1202-At542-S

## Data Availability

The data are still being analyzed for additional insight and potential publications, implying that we would prefer them not to be yet widely available.

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
