# Peer review of "A Reliable Approach for Revealing Molecular Targets in Secondary Ion Mass Spectrometry"

_ijms, 2022, doi:10.3390/ijms23094615_

Round 1

Reviewer 1 Report

The systematic study using different size of gold particle labelling for nanoSIMS is useful for the observation of biological targets. Please check the following minor additions.

Please write non abbreviation word at the first appearance. Or please make abbreviation list.

Line 81   HEK

Line 117   TOMM20

Line 118   GM130

Line 119  PMP70

In Fig.3   BSA  write here from line 370

Line 190  Cyanine 3 (Cy3)

Line 306  LA-ICPMS

Table 1  LR

Table 2  IgG

Line 334  DMEM

Line 348   PFA

Line 351   CM, PBS

Line 444  NIS

Please write the Au size used in Figure 4 and 5.

Please write using subscript.

Line 322  OsO4

Line 392  CaCl2, MgCl2

Please write the conclusion to understand clearly.

Author Response

 Response to Reviewer 1 Comments

Point 1: Please write non abbreviation word at the first appearance. Or please make abbreviation list.

Line 81   HEK

Line 117   TOMM20

Line 118   GM130

Line 119  PMP70

In Fig.3   BSA  write here from line 370

Line 190  Cyanine 3 (Cy3)

Line 306  LA-ICPMS

Table 1  LR

Table 2  IgG

Line 334  DMEM

Line 348 PFA

Line 351 CM, PBS

Line 444 NIS

Response 1: We would like to thank the reviewer for a detailed listing of the abbreviations that need to be used properly. We have checked all the above mentioned abbreviations and added the full terminology at the places where they were first mentioned when necessary. Exceptions are LR, CM and NIS. These abbreviations refer to products, devices or applications, and are commonly used directly in such simplified form. Moreover, we have replaced the HEK cell with human embryonic kidney (HEK293) cell throughout the manuscript for consistency.

Point 2: Please write the Au size used in Figure 4 and 5.

 Response 2: in Figures 4 and 5, the particle size (6 nm) is added.

Point 3: Please write using subscript.

Line 322  OsO4

Line 392  CaCl2, MgCl2

 Response 3: The corresponding compounds were updated, with the proper subscript written.

Point 4: please write the conclusion to understand clearly.

Response 4: Based on the suggestion of the reviewer, we tried to improve the discussion part and made changes in the revised manuscript to improve the clarity and organization of the discussion and conclusion. Among other revisions made, for instance, the studies mentioned are clearly indicated and the figures are referred to in corresponding parts, to help the interpretation.

Reviewer 2 Report

Li Fengxia et al present a thorough systematic study on how different gold nanoparticles (of varying size) provide different performances of nano secondary ion mass spectrometry (nanoSIMS). Notably, NP with 6nm allowed to attain an unprecedented resolution thus realizing nanoSIMS potential to label a broad range of biological targets with a high specificity in a broad array of samples/surroundings – thus overcoming one of the major drawbacks of the technique. This work is certainly fitting the Special Issue on Nano-Materials and Methods 3.0.

The conclusions are solid, the manuscript is well written and the findings are novel and fitting. Thus I fully support its publication.

One very minor comment…
In Figure 1, please rescale the size of the gold nanoparticles so they are in scale with respect to the IgG domains. For instance, the nanoparticle of 15nm is as large as the whole protein, and not only the FC domain as one could expect from the image (although not restricted to, in table 2 of the paper in nanoscale, 2016,8, 13463-13475 you can find IgG dimensions both measured by AFM/MD). As a rule of thumb, IgG FC domain is about 6nm.

Author Response

Response to Reviewer 2 Comments

Point 1: In Figure 1, please rescale the size of the gold nanoparticles so they are in scale with respect to the IgG domains. For instance, the nanoparticle of 15nm is as large as the whole protein, and not only the FC domain as one could expect from the image (although not restricted to, in table 2 of the paper in nanoscale, 2016,8, 13463-13475 you can find IgG dimensions both measured by AFM/MD). As a rule of thumb, IgG FC domain is about 6nm..

Response 1: We would like to thank the reviewer for providing this very valuable suggestion for improving the figure representation. In the revised version, we have replaced the original Figure 1 with a new version that more precisely represents the scale of protein size to nanoparticle size. Such a modification also helps the interpretation of the result in the discussion, by referring to the relative size of the probes in Figure 1.